# SARS-CoV-2: Possible recombination and emergence of potentially more virulent strains

Dania Haddad[1], Sumi Elsa John[1], Anwar Mohammad[2], Maha M. Hammad[2], Prashantha Hebbar[1], Arshad Channanath[1], Rasheeba Nizam[1], Sarah Al-Qabandi[3], Ashraf Al Madhoun[1], Abdullah Alshukry[4], Hamad Ali[1,5], Thangavel Alphonse Thanaraj[1]*, Fahd Al-Mulla[1]*

1 Department of Genetics and Bioinformatics, Dasman Diabetes Institute, Kuwait, Kuwait, 2 Department of Biochemistry and Molecular Biology, Dasman Diabetes Institute, Kuwait, Kuwait, 3 Public Health Laboratory, Ministry of Health, Kuwait, Kuwait, 4 Department of Otolaryngology & Head and Neck Surgery, Jaber Al-Ahmad Hospital, Ministry of Health, Kuwait, Kuwait, 5 Faculty of Allied Health Sciences, Department of Medical Laboratory Sciences, Health Sciences Center, Kuwait University, Kuwait, Kuwait

☯ These authors contributed equally to this work.
* alphonse.thangavel@dasmaninstitute.org (TAT); fahd.almulla@dasmaninstitute.org (FAM)

**Data Availability Statement:** All relevant data are within the paper and its Supporting Information files.

## Abstract

COVID-19 is challenging healthcare preparedness, world economies, and livelihoods. The infection and death rates associated with this pandemic are strikingly variable in different countries. To elucidate this discrepancy, we analyzed 2431 early spread SARS-CoV-2 sequences from GISAID. We estimated continental-wise admixture proportions, assessed haplotype block estimation, and tested for the presence or absence of strains' recombination. Herein, we identified 1010 unique missense mutations and seven different SARS-CoV-2 clusters. In samples from Asia, a small haplotype block was identified, whereas samples from Europe and North America harbored large and different haplotype blocks with nonsynonymous variants. Variant frequency and linkage disequilibrium varied among continents, especially in North America. Recombination between different strains was only observed in North American and European sequences. In addition, we structurally modelled the two most common mutations, Spike_D614G and Nsp12_P314L, which suggested that these linked mutations may enhance viral entry and replication, respectively. Overall, we propose that genomic recombination between different strains may contribute to SARS-CoV-2 virulence and COVID-19 severity and may present additional challenges for current treatment regimens and countermeasures. Furthermore, our study provides a possible explanation for the substantial second wave of COVID-19 presented with higher infection and death rates in many countries.

## Introduction

The severe acute respiratory syndrome coronavirus-2 (SARS-CoV-2) outbreaks have grievously impacted the world in a short span of time. Understanding the factors that govern the

**Funding:** This work was supported by Coronavirus emergency resilience grant from Kuwait Foundation for the Advancement of Sciences.

**Competing interests:** The authors have declared that no competing interests exist.

severity of a pandemic is of paramount importance to design better surveillance systems and control policies [1]. In the case of COVID-19, three variables play a critical role in its spread and morbidity in a country: the nature of the pathogen, the genetic diversity of the host population, and the environmental factors such as public awareness and governmental health measures [2].

SARS-CoV-2 spike (S) glycoprotein plays an integral role in the viral transmission and virulence [3]. The S protein contains two functional subunits, S1 and S2, cleaved by furin protease at the host cell [4]. The S1 subunit contains the receptor binding domain and facilitates the interactions with the host cell surface receptor, Angiotensin-converting enzyme 2 (ACE2) [5,6]. The S2 subunit, activated by the host Transmembrane Serine Protease 2, harbors necessary elements for membrane fusion [7]. Mutations in the S protein may induce conformational changes leading to increased pathogenicity [8]. We were the first to report the major role of D614G (23403A>G) mutation located at the S1-S2 proximal junction. This mutation generates conformational changes in the protein structure rendering the furin cleavage site (664-RRAR-667) more flexible and thus enhancing viral entry [9]. Currently, the D614G mutation has become the focus of several studies addressing prospective drug targeting strategies [10]. Furthermore, studies on understanding the genetic diversity and evolution of SARS-CoV-2 are emerging [11]. Admixture analyses have been conducted to understand the evolution of betacoronaviruses and in particular the diversification of SARS-CoV-2 [12]. Haplotype analyses have also indicated that frequencies of certain haplotypes correlate with viral pathogenicity [13].

Here we extended our previous studies [9,14] by analyzing the population genetics aspects within genome sequences of SARS-CoV-2 to understand the contiguous spread of SARS-CoV-2, its rapid evolution, and the differential severity of COVID-19 among different continents. We analyzed 2341 full-length viral sequences deposited in GISAID including those sequenced at our institute from patients in Kuwait. We found evidence of coinfection between different viral strains in Europe and in North America but not in the other continents. We also modelled two major mutations and their possible effects on the stability of the encoded protein and thereby on SARS-CoV-2 virulence.

## Methods

### Retrieval of complete SARS-CoV-2 genome sequences

2431 complete SARS-CoV-2 genome sequences from infected individuals were retrieved from the GISAID database (Global Initiative on Sharing All Influenza Data) [15] (accessed on April 3rd, 2020) and were used in all analyses.

### Alignment and annotation of amino acid sequence variation

Multiple sequence alignment was performed using MAFFT v7.407 [16] (retree: 5, maxiter: 1000). Alignment gaps were trimmed using TrimAL (automated1) [17]. SeqKit [18] was used to concatenate chopped-off sequences using NC045512.2 as the reference genome sequence. SNP-sites [19] and Annovar [20] were used to extract and annotate single nucleotide variants (SNV).

### Linkage disequilibrium and haplotype blocks analysis

PLINK2 [21] was used to extract sequence variants with minor allele frequencies (MAF) $\geq$ 0.5%, to estimate inter-chromosomal linkage disequilibrium (LD), squared correlation coefficient ($r^2$), and haplotype blocks. We used Haploview [22] to visualize the haplotype blocks. We

examined the correlation between the LD results and physical distance across all continental data sets.

To detect recombination within datasets, we tested individual datasets for pairwise homoplasy index using PhiPack software [23]. All the PhiPack tests were performed with 1000 permutations. We used 'Profile' program available in PhiPack suite with default options for window scan size and step size in all the aligned sequence datasets. Further validation was done using RDP, GENECONV, MaxChi², Chimaera, and 3Seq algorithms available from RDP4 software suite [24] with default parameter settings and 1000 permutations.

Admixture 1.3.0 software [25] was used to identify genetic substructure of strains across the continental transmission as follows: variants with MAF $\geq$ 0.5%, variants in LD ($R^2 > 0.5$), haplotype blocks, independent variants, nonsynonymous, and synonymous variants. All analyses were iterated for K = 20 and cross validation errors were examined to infer an optimal K cluster. Replicate runs were further processed using CLUMPAK [26] and results for the major modes were illustrated using the ggplot2 data visualization package for the statistical programming language R (https://www.r-project.org/).

## Protein structural analysis

The crystal structures of the viral RNA-dependent RNA polymerase (RdRp, PDB ID: 6M71) [27] and the S protein (PDB ID:6VSB) [28] were used as a scaffold to model the amino acid variations observed in the viral strains. The missing amino acids, i.e. invisible in the Cryo-EM structure of the S protein, were modelled-in by using SWISS-Model [29]. DynaMut web server [30] was used to predict the effect of the mutations on the proteins stability and flexibility. PyMol (Molecular Graphics System, Version 2.0, Schrodinger, LLC) was used to generate structural images.

## Results

### Detection and classification of mutations from global SARS-CoV-2 genome sequences

We analyzed 2431 high quality SARS-CoV-2 genome sequences from six continental groups. 2352 sequences showed substantial genetic differences from the Wuhan reference sequence (NC045512.2). We identified 1010 unique mutations using our variant calling pipeline. 613 variants were nonsynonymous, 387 variants were synonymous, 9 variants were stop-gain, and 1 variant in the 5′ UTR. We found only 72 variants with MAF $\geq$ 0.5%, which were used in admixture and haplotype block analyses. **Fig 1** shows the distribution of synonymous and nonsynonymous variants in each gene of the SARS-CoV-2 genome with varying MAF thresholds. The genes with the highest percentage of nonsynonymous variants and MAF $\geq$ 0.5% were: ORF3a, M, and ORF8, while the genes with the highest percentage of synonymous variants and MAF $\geq$ 0.5% were: ORF6 and ORF10.

### Identification of SARS-CoV-2 genetic clusters in different continents

The principal components analysis (PCA) of early spread 2352 SARS-CoV-2 sequences gave three distinct clusters of samples based on their continent of origin (**Fig 2**). All three clusters diverged from a single point (**Fig 2**, red circle). The North American cluster showed the least viral genetic variances unlike the samples from Asia and Oceania which harbored the most genetic diversity. The European cluster is well-defined with few interspersed Asian samples, which is an indication of its origin. This clustering in Europe and North America is probably associated with a founder effect where a single mutation was introduced and subsequently

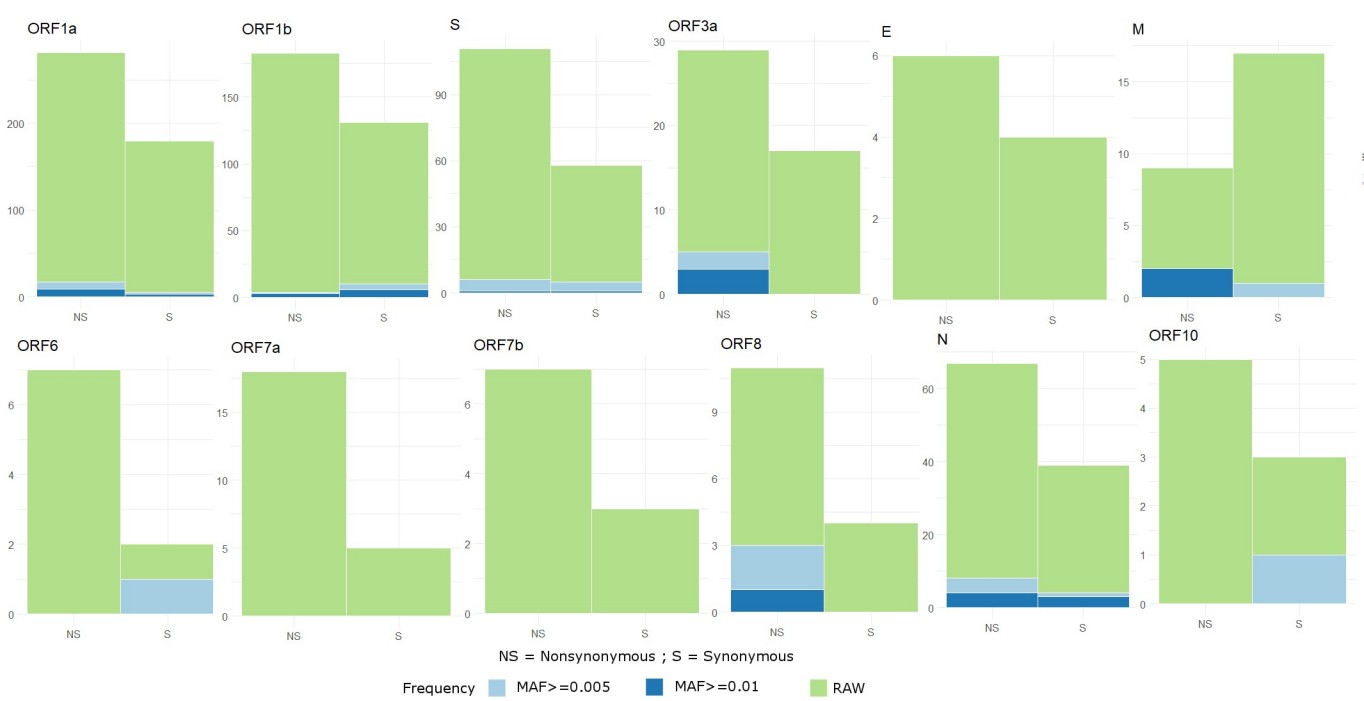

**Fig 1. Detection and classification of mutations from GISAID SARS-CoV-2 genome sequences.** Illustration of the distribution of synonymous and nonsynonymous variants for each gene in the raw dataset, MAF $\geq$ 0.5%, and MAF $\geq$ 1% thresholds are shown. A set of 72 variants in total was observed with MAF $\geq$ 0.5% threshold and utilized in subsequent analysis. MAF- Minor Allele Frequency.

transmitted. This suggestion is corroborated by the fact that the collection date of the founder strain is prior to those in the European and the North American clusters (**S1 Fig**).

The admixture analysis showed a gradual reduction in cross validation (CV) error for iterations up to K = 7 (**S2 Fig**). Subsequent iterations showed a fluctuating pattern; however, the least error was seen at K = 17. Further, we verified this CV trend in a subset of variants filtered for MAF $\geq$ 0.5% (comprised of 72 SNVs). Interestingly, we again observed a gradual reduction in CV error up to K = 7 (with best fit at 7) and subsequently, a trend of increasing CV error.

We created subsets of strong LD, weak LD, Haplotype block, nonsynonymous, and synonymous variants from 72 variants at K = 7 across the continents. We separately performed admixture analysis on these subsets. The analysis of the detected seven datasets revealed interesting mosaic patterns (**Fig 3**A and **3**B). Samples from Asia formed largely two clusters (C1and C6 with C1 as dominant); whereas, samples from the European dataset were distributed into six different clusters (C1, C2, C3, C5, C6 and C7 with C2 as dominant); and the North American samples formed four clusters (C1, C2, C4 and C7 with C4 as dominant). The African and Oceanian datasets formed two clusters each ([C2 and C3 with C2 as dominant] and [C3 andC5 with C3 as dominant cluster], respectively) and South American dataset was formed by five clusters (C1, C2, C3, C5 and C6).

In the context of dependent variants (**Fig 3**C and **3**D), a strong LD block ($R^2 \geq 0.5$) was mostly observed in two clusters (C1 and C6) and a weak LD block ($R^2 < 0.5$) was observed in three clusters (C1, C3, and C7) in Asia. In Europe, a strong LD block was observed in 4 clusters (C1, C2, C3, and C5) and a weak LD block was observed in five clusters (C1, C2, C3, C4, and C5; predominantly from C3 and C5). Interestingly, four clusters were common between the strong and weak LD blocks, suggesting that a makeup of four strains that dominate in Europe have significant proportion of strong and weak LD signatures in them. Likewise, in North America, both strong and weak LD were observed among three clusters (C1, C2, and C4) and (C1, C3, and C4)

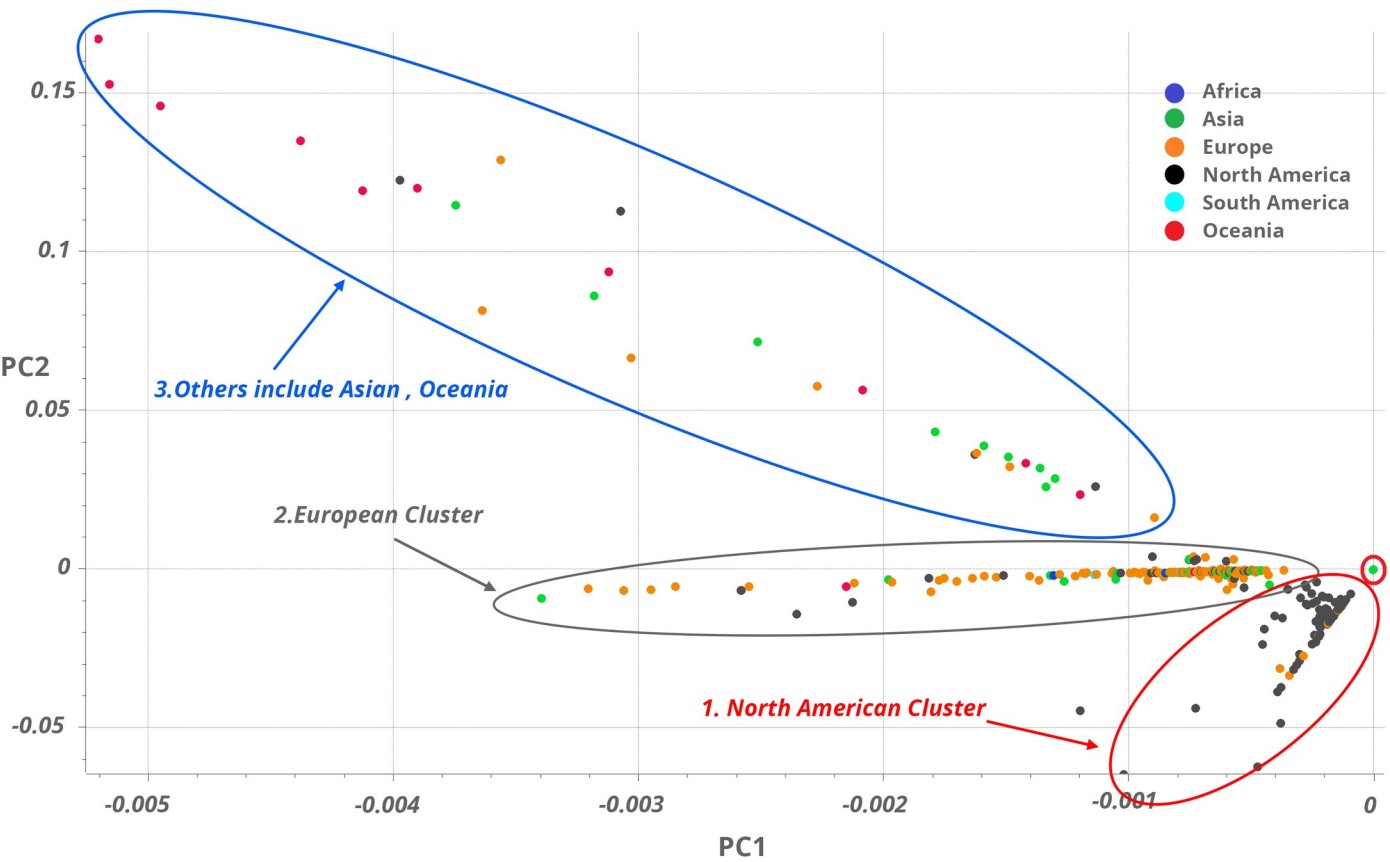

**Fig 2. Principal Component Analysis using 2352 GISAID sequences.** Principal Component Analysis of 2352 SARS-CoV-2 sequences shows three distinct clusters of color-coded samples (see the legend for their continent of origin). All three clusters diverge from a single point (red circle). The North American cluster (black oval) shows least variance among the three. The European cluster (orange oval) is well-defined with few interspersed Asian samples, an indication of its origin. The third cluster (Blue oval) shows the most variance and includes samples from Oceania, Asia, and others.

respectively. Further in Africa, Oceania, and South America, strong LD was observed among (C2), (C1) and (C1 and C2) respectively; weak LD was observed in (C3 and C4), (C1, C2, C3, and C5) and (C3 and C5) respectively. Interestingly, proportions of haplotype blocks (**Fig 3**E), identified using the whole data, followed mostly the pattern observed in the track of strong LD (i.e. the track of **Fig 3**C) but admixed with weak LD strains of the respective continents.

Of particular note, variations in proportions of strains carrying nonsynonymous and synonymous signatures were also very evident (**Fig 3**F and 3G). Proportions of two nonsynonymous clusters (C1 and C5, which have admixed with each other) and two synonymous clusters were dominating in Asia. Four nonsynonymous clusters (C2, C4, C5, and C6) and five synonymous clusters were dominating in Europe, while in North America, three nonsynonymous (C1, C3, C6) and two synonymous (C1, C3) clusters were in higher proportions. Similarly, C4 and C6, C5, and C2 nonsynonymous clusters along with C1 and C6, C2, and C5 synonymous clusters were in high proportions in Africa, Oceania, and South America respectively.

## Evidence of coinfection in the sequences from continental datasets

Next, we tested for the presence or absence of recombination among continental datasets using PhiPack software, which uses three different tests, namely "Pairwise Homoplasy Index (Phi)" [23], "Neighbor Similarity Score (NSS)" [31], and "Maximum $\chi^2$ (MaxChi$^2$)"[32].

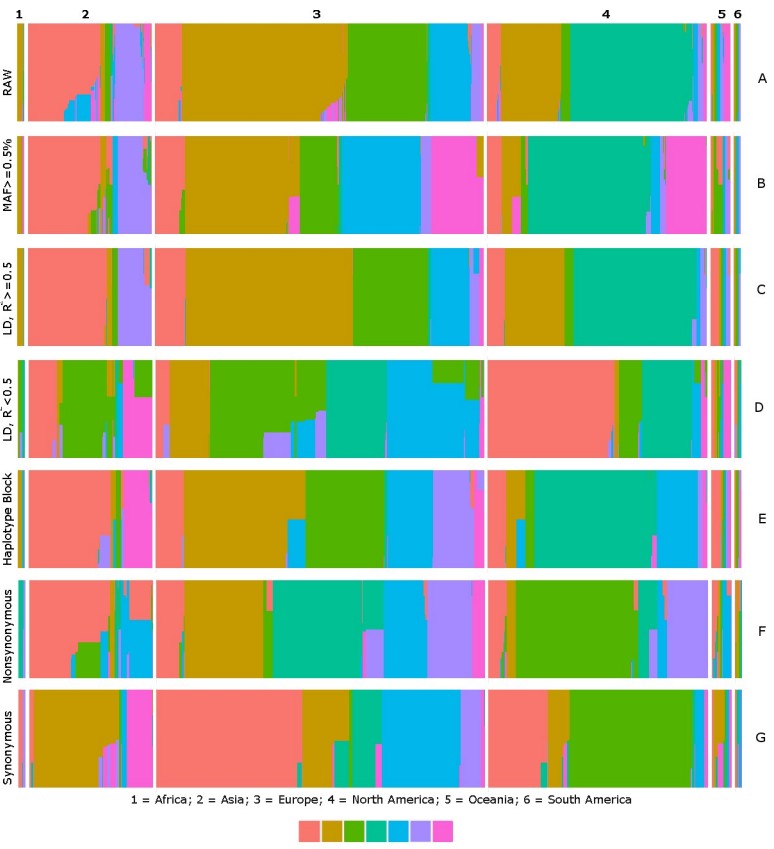

**Fig 3. Identification of SARS-CoV-2 genetic clusters in different continents.** Illustration of the seven (C1 to C7, color-coded) genetic subdivisions of SARS-CoV-2 sequences across continents using variants with MAF ≥ 0.5%. Differential proportions of strong LD (C), weak LD (D), haplotype block (E), nonsynonymous (F), and synonymous (G) variants across continental datasets are shown.

The results from these tests on the combined dataset suggested the possibility of coinfection at a global level (**Table 1**). However, significant P-values (< 0.05) for Phi statistics were observed only with European and North American populations: European (NSS test, P-value = 0.001) and North American (NSS and Phi (normal), P-value = 0.007, 0.042, respectively). These significant P-values showed evidence of the presence of recombination events on a continental level *i.e.* European and North American populations. These two populations gave the highest numbers of informative variants, 276 and 194 respectively. Furthermore, a relatively large number of regions with significant evidence of recombination was observed in these two continents (**Table 1**). Moreover, plausible evidence of recombination in North American and European datasets were ascertained using algorithms available from RDP4 software: recombination events in European population were confirmed with significant P-values by the MaxChi$^2$, Chimera, and 3Seq algorithms while possible recombination events in North American populations were validated by MaxChi$^2$ and 3Seq algorithms (**Table 2**). On the flip side; African, Oceanic, South American, and Asian datasets showed no recombination in early spread of SARS-CoV-2 virus in the respective continents.

## Estimation of LD and haplotype blocks in continental samples

Haplotype block analysis was carried out using two approaches; first, because of the small sample size in African (n = 25), Oceanian (n = 69), and South American (n = 24) datasets, we

**Table 1. Evidence of recombination in the sequences from continental datasets and regions exhibiting evidence of recombination observed from 'Profile' program in continental datasets.**

| Dataset | Number of informative variants | Regions showing significant (p<0.05) evidence of recombination | Tests to detect evidence of recombination | Significance of observed Phi statistics |
|---|---|---|---|---|
| Africa (n = 25) | 19 | 18800–18900, 22450–23425, 29300–29325 | NSS | 1 |
| | | | MaxChi$^2$ | 0.978 |
| | | | Phi (permutation) | 1 |
| | | | Phi (normal) | 1 |
| Asia (n = 364) | 127 | 4600–4825, 5200–5825, 6425–6500, 6900–7375, 8800–8925, 9175–9750, 9950–10150, 21000–21050, 22650–23625 | NSS | 0.113 |
| | | | MaxChi$^2$ | 0 |
| | | | Phi (permutation) | 0.493 |
| | | | Phi (normal) | 0.305 |
| Europe (n = 1132) | 276 | 500–700, 2100–2125, 3300–3325, 4050–4725, 5150–5450, 7825–7850, 12925–14075, 18825–18850, 19375–19400, 20075–20475, 22975–23000, 23575–23625, 25275–25325, 26275–26325, 29200–29325 | **NSS** | **0.001** |
| | | | MaxChi$^2$ | 0.208 |
| | | | Phi (permutation) | 0.343 |
| | | | Phi (normal) | 0.223 |
| North America (n = 738) | 194 | 800–1100, 5575–6300, 6475–6525, 7300–7450, 18800–18875, 19950–19975, 21525–22625, 23500–23600, 24275–24675, 24975–25000, 25725–26050, 26375–26425, 27600–27800, 29300–29325 | **NSS** | **0.007** |
| | | | MaxChi$^2$ | 0.061 |
| | | | Phi (permutation) | 0.060 |
| | | | **Phi (normal)** | **0.042** |
| South America (n = 24) | 24 | None | NSS | 0.596 |
| | | | MaxChi$^2$ | 0.680 |
| | | | Phi (permutation) | 0.717 |
| | | | Phi (normal) | 0.454 |
| Oceania (n = 69) | 50 | 22625–23425 | NSS | 1 |
| | | | MaxChi$^2$ | 0.502 |
| | | | Phi (permutation) | 0.872 |
| | | | Phi (normal) | 0.361 |
| Combined (n = 2352) | 554 | 1625–1650, 3300–3525, 3975–4225, 4625–4725, 5175–5375, 7550–7800, 8800–8925, 9175–9750, 9925–10050, 10600–12125, 13025–14075, 18175–18550, 18850–19150, 20075–20475, 23000–23025, 23475–23525, 24700–24750, 27050–28225 | **NSS** | **0.003** |
| | | | **MaxChi$^2$** | **0.001** |
| | | | **Phi (permutation)** | **0.008** |
| | | | **Phi (normal)** | **0.015** |

Results of NSS, MaxChi$^2$, Phi (permutation) and Phi (normal) tests using pairwise homoplasy index test available from PhiPack software on the combined dataset of all the 2352 samples. Significant P-values suggest the possibility of coinfection on a global level. European (NSS test, P-value of 0.001) and North American (NSS and Phi (normal), P-value of 0.007, 0.042 respectively) show evidence for the presence of recombination events, while African, Oceanic, South American, and Asian datasets show no recombination in early spread of SARS-CoV-2 in respective continents.

decided to compare haplotype blocks obtained from pooled datasets of all variants with MAF ≥ 0.5%. Hence, we initially compared LD blocks obtained from the combined dataset and then observed LD among the same variants in each continental dataset. In the second approach, we estimated haplotype blocks only within continental datasets with large sample size such as Asia, Europe, and North America.

The first analysis, using the combined dataset, showed that LD block varies among continental datasets. **S3 Fig** illustrates the extent of LD variation in haplotype block of the combined pool in each continental dataset. Examination of the variants in haplotype blocks suggested a clear variation in allele frequency between continental datasets. **Table 3** shows MAF of 18 variants involved in haplotype block of the combined data in each continental dataset. These differences called for the second approach of estimating haplotype blocks and the extent of LD

**Table 2. Plausible recombination events validated by RDP4 suite.**

| Continental datasets | Event | Start | End | RDP | GENECONV | MaxChi$^2$ | Chimera | 3Seq |
|---|---|---|---|---|---|---|---|---|
| North American | 1~ | 530* | 29326 | NS | NS | 3.84E-03 | NS | 1.14E-04 |
| | 2~ | 226* | 29535 | NS | NS | 1.05E-02 | NS | 2.54E-04 |
| | 3~ | 175* | 26046 | NS | NS | 1.02E-02 | NS | 4.83E-03 |
| European | 1~ | 820* | 24825* | NS | NS | 9.72E-05 | 2.12E-03 | 4.47E-04 |
| Combined | 1~ | 1519* | 29004 | NS | NS | 1.09E-03 | NS | 1.07E-04 |
| | 2~ | 29575* | 29835* | NS | 4.98E-05 | NS | NS | 2.56E-02 |

Analysis performed using RDP, GENECONV, MaxChi$^2$, Chimera, and 3Seq algorithms.

NS = No significant p-value is observed for the recombinant event using respective method.

* = The actual breakpoint position is undetermined (it was most likely overprinted by a subsequent recombination event.

~ = It is possible that this apparent recombination signal could have been caused by an evolutionary process other than recombination.

between variants directly from individual continental datasets having large sample size. Surprisingly, we observed different sets of variants in haplotype blocks, different lengths of haplotype blocks, and differences in nonsynonymous composition in haplotype blocks among the three continents datasets (Fig 4). Table 4 describes characteristics of the haplotype blocks observed in the datasets from Asia, Europe, and North America. Correlation between LD and physical distance suggested that a strong LD was observed in North American sequences throughout the genome compared to sequences from other continents (Fig 5). The South American and European sequences showed strong LD towards the upstream region of ORF1a gene, while North American and Oceanic sequences showed strong LD from the S protein region to the end of the genome.

**Table 3. MAF distribution of 18 variants involved in haplotype block of combined dataset in each continental data.**

| SNV | Minor allele frequency | | | | | | | Functional consequence | Gene |
|---|---|---|---|---|---|---|---|---|---|
| | Africa | Asia | Europe | North America | South America | Oceania | Combined | | |
| 241CT | 0.08 | 0.0811 | 0.2507 | 0.3231 | 0.4583 | 0.1905 | 0.49 | downstream | 5'-UTR |
| 1059CT | 0.24 | 0.019 | 0.1435 | 0.1865 | 0 | 0.0289 | 0.1313 | nonsynonymous | ORF1a |
| 3037CT | 0.08 | 0.0760 | 0.2502 | 0.313 | 0.4583 | 0.1884 | 0.4804 | synonymous | ORF1a |
| 8782CT | 0.04 | 0.2304 | 0.0424 | 0.4197 | 0.2917 | 0.1884 | 0.2484 | synonymous | ORF1a |
| 11083GT | 0.04 | 0.269 | 0.1339 | 0.0531 | 0.125 | 0.4928 | 0.1416 | nonsynonymous | ORF1a |
| 14408CT | 0.08 | 0.0760 | 0.25 | 0.3148 | 0.4583 | 0.1884 | 0.481 | nonsynonymous | ORF1b |
| 14805CT | 0.08 | 0.0047 | 0.1366 | 0.0224 | 0.3333 | 0.1159 | 0.0787 | synonymous | ORF1b |
| 17747CT | 0 | 0 | 0.0106 | 0.4616 | 0 | 0.0579 | 0.174 | nonsynonymous | ORF1b |
| 17858AG | 0 | 0 | 0.0097 | 0.4418 | 0 | 0.0579 | 0.1798 | nonsynonymous | ORF1b |
| 18060CT | 0 | 0.0166 | 0.0088 | 0.4382 | 0 | 0.0579 | 0.1829 | synonymous | ORF1b |
| 23403AG | 0.08 | 0.0783 | 0.25 | 0.3121 | 0.4583 | 0.1884 | 0.4808 | nonsynonymous | S |
| 25563GT | 0.32 | 0.0213 | 0.1851 | 0.2262 | 0 | 0.0434 | 0.1647 | nonsynonymous | ORF3a |
| 26144GT | 0.04 | 0.1119 | 0.1293 | 0.0211 | 0.125 | 0.1594 | 0.0923 | nonsynonymous | ORF3a |
| 27046CT | 0 | 0 | 0.1114 | 0.0013 | 0.125 | 0.0144 | 0.0538 | nonsynonymous | M |
| 28144TC | 0.04 | 0.2304 | 0.0407 | 0.4185 | 0.2917 | 0.1884 | 0.2483 | nonsynonymous | N |
| 28881GA | 0 | 0.0381 | 0.2396 | 0.0423 | 0.375 | 0.0869 | 0.1378 | nonsynonymous | N |
| 28882GA | 0 | 0.0381 | 0.2396 | 0.0410 | 0.375 | 0.0869 | 0.1374 | synonymous | N |
| 28883GC | 0 | 0.0381 | 0.2396 | 0.0410 | 0.375 | 0.0869 | 0.1374 | nonsynonymous | N |

Display of minor allele frequency for each variant in different continents, the functional consequence of these variants, and their corresponding genes. (SNV- single nucleotide variant).

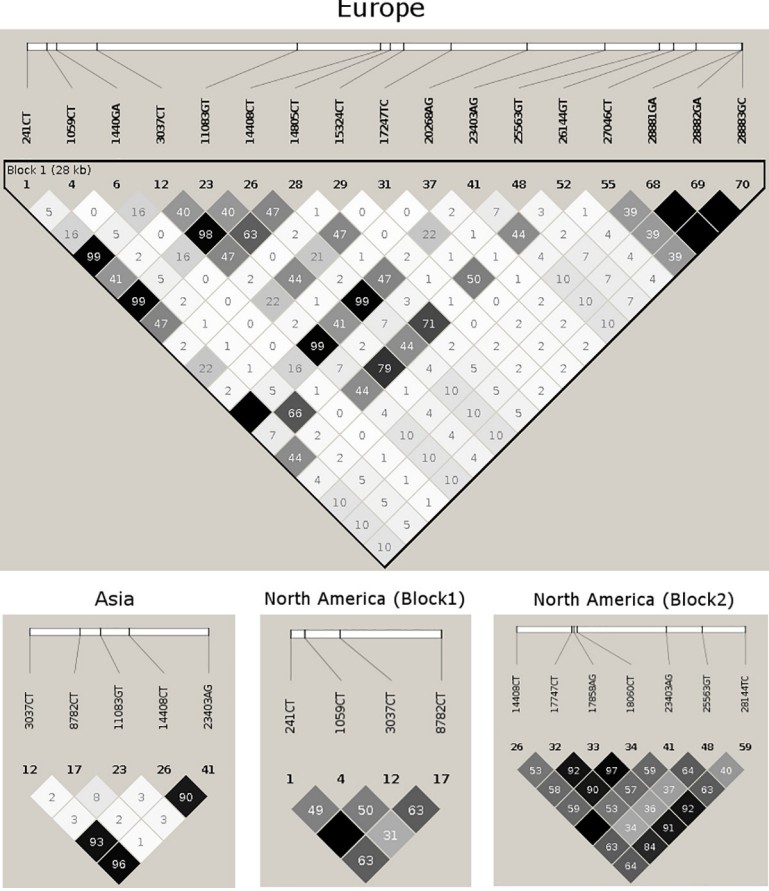

**Fig 4. Estimation of haplotype blocks in continental samples.** Haplotype block estimation and extent of linkage disequilibrium observed between variants in Asia, Europe, and North America, identified a single block with different lengths in Asia and Europe, while in North America two blocks were identified.

## Structural analysis of SARS-CoV-2 mutations

**D614G mutation.** Wrapp *et al.* recently solved the cryo-EM structure of the S protein with 3.5 Å resolution [28] (**Fig 6**A). The S protein has many flexible loop regions that were not visible in the structure, including the RRAR cleavage site. Therefore, we modelled-in the cleavage site and the undetected flexible regions using the cryo-EM structure PDB ID:6M71 as a scaffold [29]. **Fig 6**B shows the overlay of the S protein from PDB ID:6VSB with the modelled S protein, with an RMSD of 0.25 Å. As shown in the overlay figure, there were more loops present in the modelled structure that were undetectable by Cryo-EM. The RRAR cleavage site (**Fig 6**C) presents a highly accessible surface region, where the host protease enzyme can readily cleave the S protein [28]. As such, any mutations on the S protein, close to RRAR furin protease cleavage site, might alter its activity. Therefore, the D614G mutation is believed to increase SARS-CoV-2 virulence [9,33]. One possibility is that the change from a negatively charged aspartate to a non-polar glycine may modify the structure and therefore the function of the protein. Charged amino acids form ionic and hydrogen bonds (H-bond) through their side chains and stabilize proteins [34]. The targeted aspartate is present in the loop region, therefore a mutation to a glycine would cause unfolding of the loop and possibly render it more flexible making the furin cleavage site more accessible.

**Table 4. Characteristics of haplotype blocks estimated from three continental datasets.**

| Dataset | Haplotype block start | Haplotype block end | Length (in kb) | Number of variants | Number of nonsynonymous variants | Variant | MAF |
|---------|----------------------|---------------------|----------------|--------------------|----------------------------------|---------|-----|
| Asia | 3037 | 23403 | 20.367 | 5 | 3 | 3037CT | 0.076 |
| | | | | | | 8782CT | 0.23 |
| | | | | | | **11083GT** | **0.269** |
| | | | | | | **14408CT** | **0.076** |
| | | | | | | **23403AG** | **0.078** |
| Europe | 241 | 28883 | 28.643 | 17 | 10 | 241CT | 0.25 |
| | | | | | | **1059CT** | **0.143** |
| | | | | | | **1440GA** | **0.052** |
| | | | | | | 3037CT | 0.25 |
| | | | | | | **11083GT** | **0.134** |
| | | | | | | **14408CT** | **0.25** |
| | | | | | | 14805CT | 0.136 |
| | | | | | | 15324CT | 0.062 |
| | | | | | | 17247TC | 0.0689 |
| | | | | | | 20268AG | 0.0734 |
| | | | | | | **23403AG** | **0.25** |
| | | | | | | **25563GT** | **0.185** |
| | | | | | | **26144GT** | **0.129** |
| | | | | | | **27046CT** | **0.111** |
| | | | | | | **28881GA** | **0.239** |
| | | | | | | 28882GA | 0.239 |
| | | | | | | **28883GC** | **0.239** |
| North America | 241 | 8782 | 8.54 | 4 | 1 | 241CT | 0.323 |
| | | | | | | **1059CT** | **0.186** |
| | | | | | | 3037CT | 0.313 |
| | | | | | | 8782CT | 0.419 |
| | 14408 | 28144 | 13.737 | 7 | 6 | **14408CT** | **0.3148** |
| | | | | | | **17747CT** | **0.462** |
| | | | | | | **17858AG** | **0.442** |
| | | | | | | 18060CT | 0.438 |
| | | | | | | **23403AG** | **0.312** |
| | | | | | | **25563GT** | **0.226** |
| | | | | | | **28144TC** | **0.418** |

Characteristics of haplotype blocks estimated from Asian, European, and North American datasets. Nonsynonymous variants are shown with bold font.

D614 is in close vicinity to T859 of the adjacent monomer's S2 (Chain B); thus, they can form a H-bond (**Fig 7**) through both sidechains. In addition, backbone H-bonds can be formed with A646 of the same chain. It was documented that S2 domains alter their structure after furin site cleavage [35]. Therefore, the mutation of D614 to G might weaken the stability of S2 and make cell entry more aggressive. It is probably the loss of the H-bond between G614 (S1/Chain A) and T859 (S2/Chain B) that stops the hinging of the S2 domain making it more flexible in the transition state when interacting with the host cell receptor. A thermodynamic analysis showed that D614G mutation resulted in slightly destabilizing the protein with a $\Delta\Delta G$: -0.086 kcal/mol and increasing the vibrational entropy to $\Delta\Delta S_{Vib}$ 0.137 kcal.mol$^{-1}$.K$^{-1}$ as seen in **Fig 8**A where the red parts indicate more flexibility. Since this mutation will occur on the trimeric structure of the S protein, all the three domains will be more flexible. Such flexibility

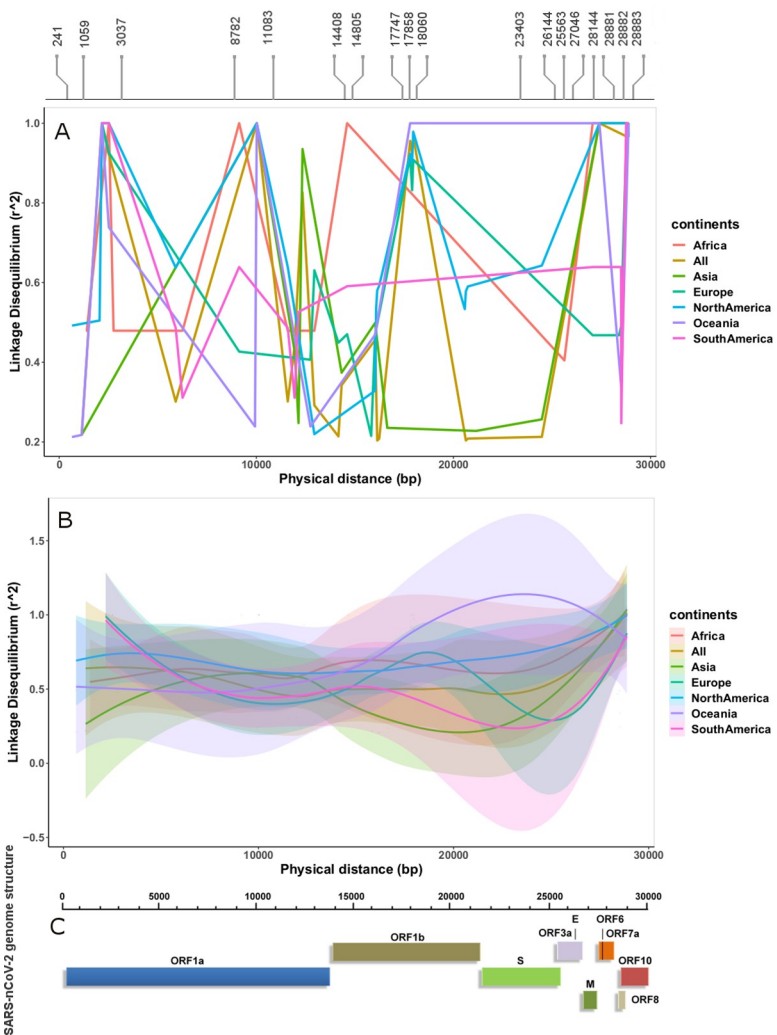

**Fig 5. Correlation between LD and physical distance across continental SARS-CoV-2 genomes.** (A) Upward peaks show strong LD and downward peaks show weak LD. North America and Oceania showed strong LD in the region overlapping the S protein. Strong LD among variants suggested that these de novo mutations were not broken by recombination events, (B) Smooth lines showed clear differences in LD over physical distance across continental datasets, (C) The genomic structure of SARS-CoV-2 is depicted.

will render the furin cleavage site more accessible which is concomitant with the virulence of the D614G mutation. Furthermore, Wrapp *et al.* suggested that the flexibility observed in the receptor binding domain region (RBD) (**Fig 8**) may facilitate the binding of ACE2 to the S protein [28].

**P314L/P323L mutation.** ORF1a and ORF1b produce a set of non-structural proteins (nsp) which assemble to facilitate viral replication and transcription (nsp7, nsp8, and nsp12) [36]. The nucleoside triphosphate (NTP) entry site and the nascent RNA strand exit paths have positively charged amino acids, are solvent accessible, and are conserved in betacoronaviruses [37] (**Fig 9**). P314L mutation (or Position P323 on the protein structure PDB ID:6M71 because of a frame shift and written as P323L hereafter) is positioned on the interface domain of the RdRp (or nsp12) between A250-R365 residues. Previous studies have shown that the interface domain has functional significance in the RdRp of *Flavivirus*. In addition, when polar or charged residue mutations were introduced into these sites, viral replication levels

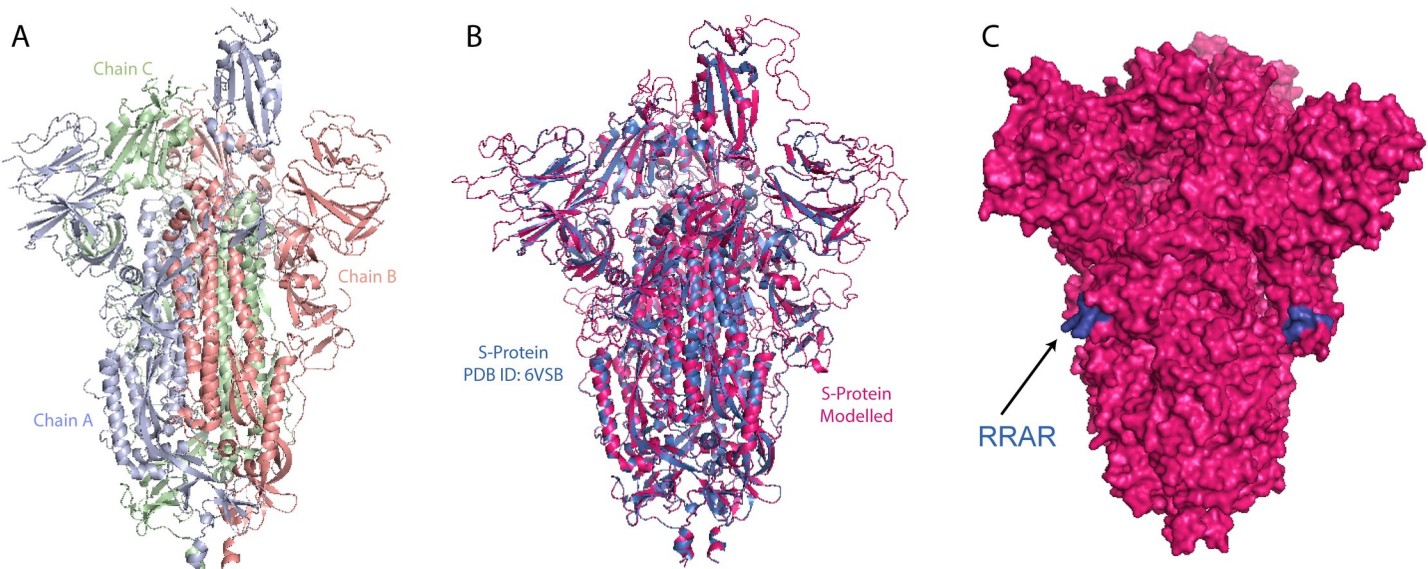

**Fig 6. 3D modelling of SARS-CoV-2 Spike protein.** (A) Trimeric structure of SARS-CoV-S spike like protein (PBD:6VSB). (B) Overlay of the SARS-CoV-S spike like protein (PBD ID: 6VSB, blue) with the modelled SARS-CoV-2 S protein (PDB ID: 6M71, magenta). (C) The surface of the modelled S protein with the RRAR furin cleavage site (blue).

were significantly affected [38]. Thus, mutations on nsp12 interface residues may affect the polymerase activity and RNA replication of SARS-CoV-2. Proline is often found in very tight turns in protein structures and can also function to introduce kinks into α-helices. In **Fig 10**, we investigated the proposed intermolecular bonds that P323 can make, where the backbone COO- group of proline can form H-bonds with the backbone NH groups of T324 and S325 or

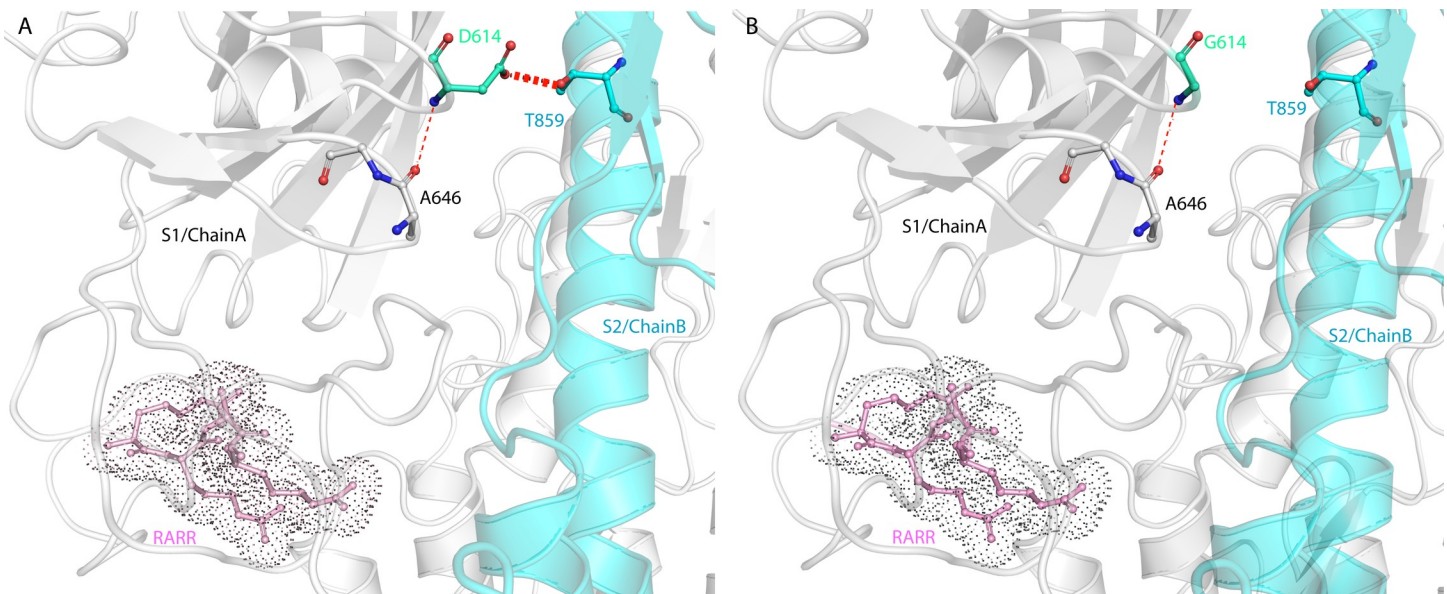

**Fig 7. 3D modelling of SARS-CoV-2 Spike protein showing suggested bonds for D614.** (A) Suggested hydrogen bonds (red dashed lines) of D614 (S1 domain chain A) with T859 (S2 domain chain B) and D614 and A646 of S1 domain chain A. (B) The suggested hydrogen bond can be disrupted with the D614G mutation altering the activity of the protein.

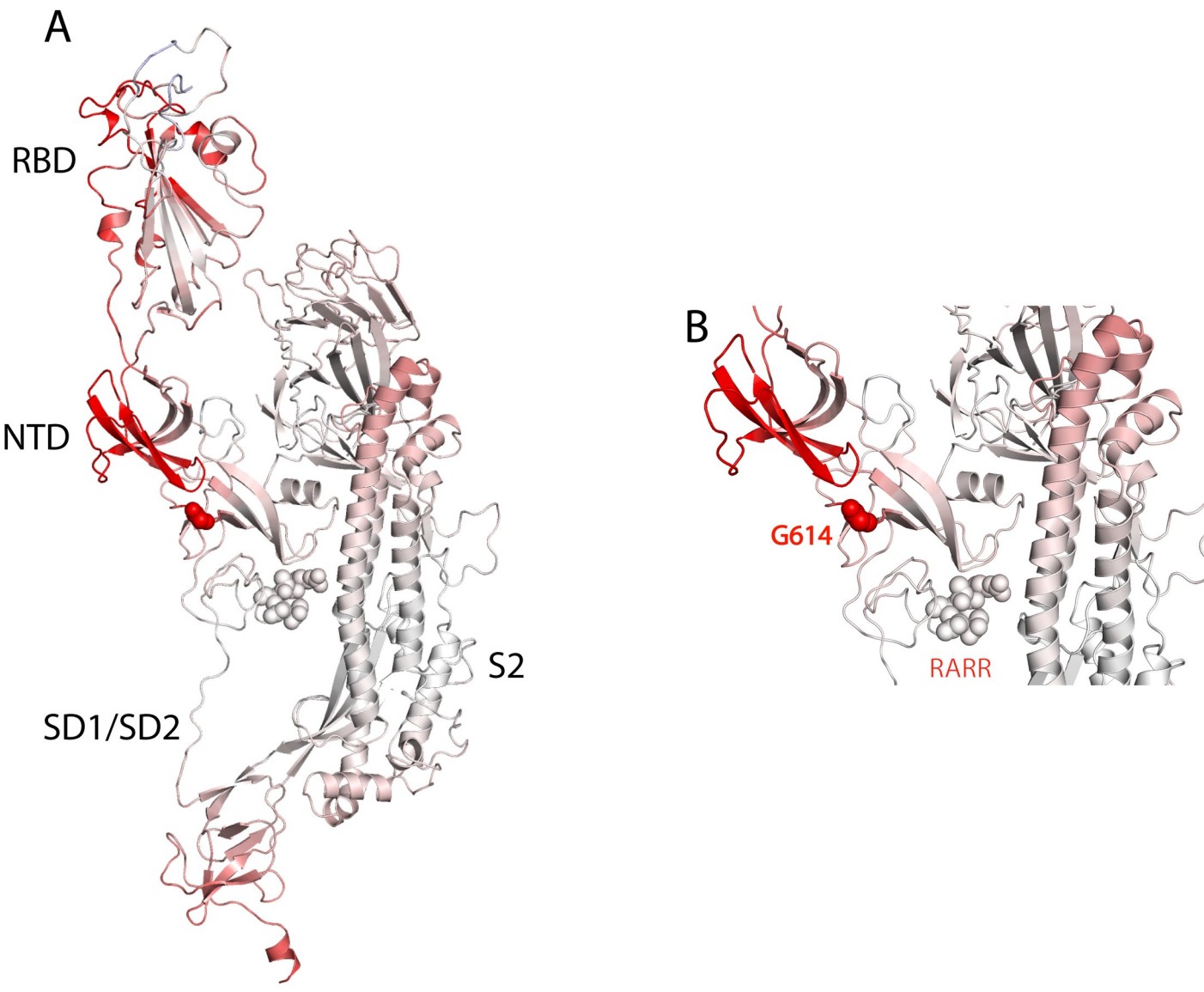

**Fig 8. 3D modelling of G614 mutation.** (A) S protein monomer 6VSB with D614G mutation, the red region of the protein depicts the more flexible region of the protein due to the D614G mutation with a decrease in stability of $\Delta\Delta G$: -0.086 kcal/mol and an increase in vibrational entropy to $\Delta\Delta S Vib$ 0.137 kcal.mol$^{-1}$.K$^{-1}$. (B) A zoomed-in structure of the N-terminal domain (NTD) and the G614 mutation in close vicinity to the RARR furin cleavage site.

the OH- group of S325 side chain. The pyrrolidine, on the other hand, forms hydrophobic interactions with W268 and F275 (**Fig 10**A). The mutation to leucine tightens the structure and reduces the flexibility with an increase in $\Delta\Delta G$: 0.717 kcal/mol and a decrease in vibrational entropy to $\Delta\Delta S_{Vib}$ ENCoM: -0.301 kcal.mol$^{-1}$.K$^{-1}$ (**Fig 11**). Furthermore, leucine possesses a non-polar side chain, seldomly involving catalysis, which can play a role in substrate recognition such as binding/recognition of hydrophobic ligands. L323 backbone COO- forms a H-bond with the sidechain OH- group of S325, in addition to a hydrophobic interaction with W68 and L270 (**Fig 10**B). L270 positioned on top of the flexible loop region, forming a hydrophobic interaction would therefore displace any water molecules entering the looped region and thereby make it more compact. The overall improved stability of RdRp can make it more efficient in RNA replication and hence increase SARS-CoV-2 replication.

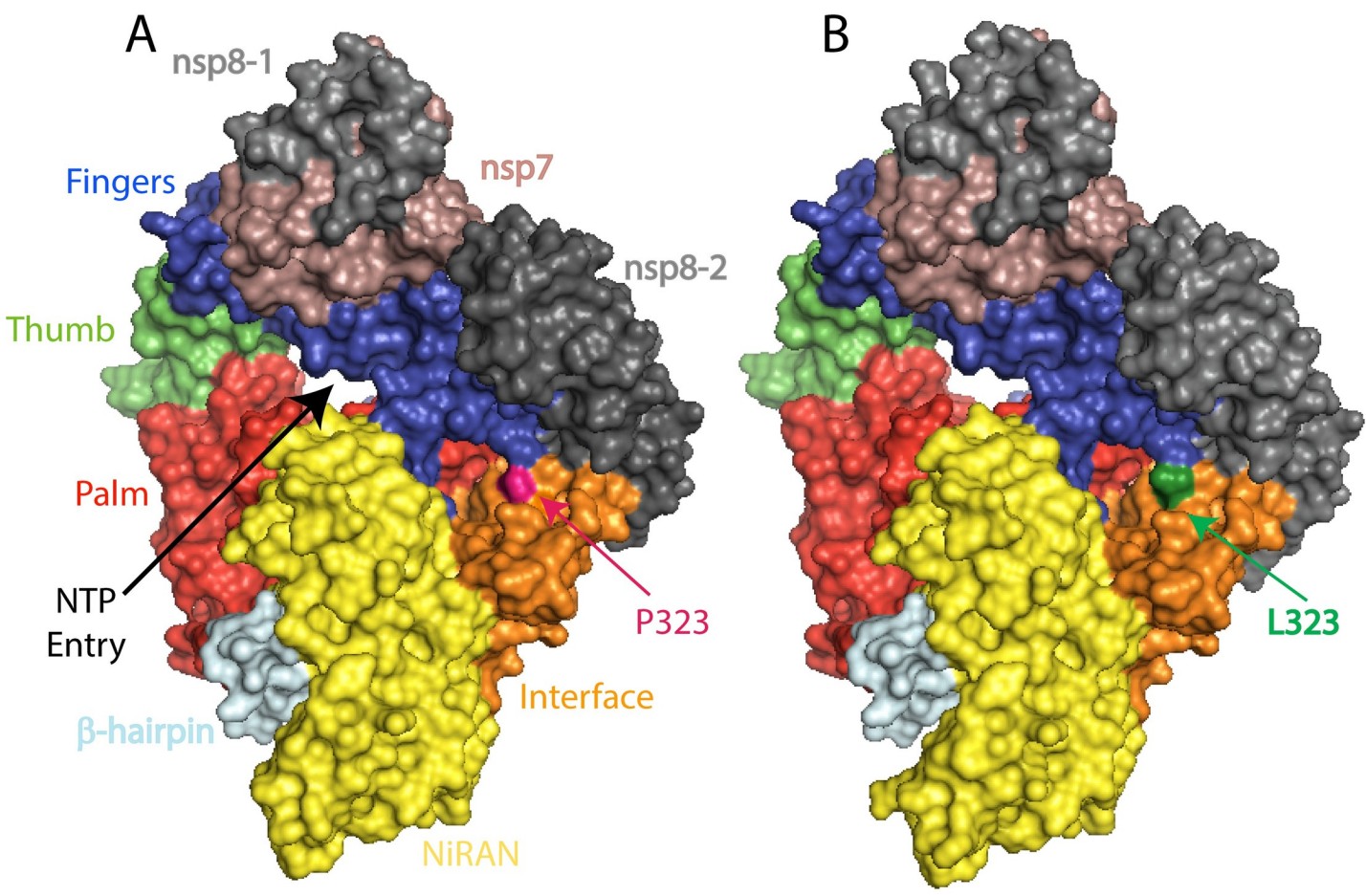

**Fig 9. SARS-CoV-2 RNA-dependent RNA polymerase structure in complex with nsp7 and two nsp8.** The viral RNA template and NTP entry is shown in black arrow heads. The active site is a large groove with several structural pockets. (A) Wild type RdRp complex P323 is shown in pink (B) L323 mutation is shown in green. RdRp-RNA-dependent RNA polymerase.

## Discussion

Our pairwise homoplasy index tests suggest that, among continental datasets, European and North American sequences have shown evidence for the presence of recombination events (P-value = 0.001 and 0.007 respectively); while African, Oceanic, South American, and Asian datasets have shown no recombination events. The recombination events in European population were further confirmed with significant P-values by MaxChi$^2$, Chimera, and 3Seq tools, while possible recombination events in North American populations were confirmed by MaxChi$^2$ and 3Seq tools. This indicates once more that European and North American populations are at higher risk of coinfection with different SARS-CoV-2 variants simultaneously. The recombination effects might also lead to deletion of big portions of RNA such as the one reported in Holland *et al*. paper, where an 81-nucleotide deletion was detected in SARS-CoV-2 ORF7a [39]. Similar deletions might decrease viral fitness and affect COVID-19 pandemic trajectory.

Depending on the variance seen within the clustered samples, PCA analysis indicated that clustering in Europe and North America is probably associated with a founder effect where a single mutation was introduced and subsequently transmitted; such founder mutations are 23403AG in Europe and 28144TC in North America. Admixture analysis identified differing

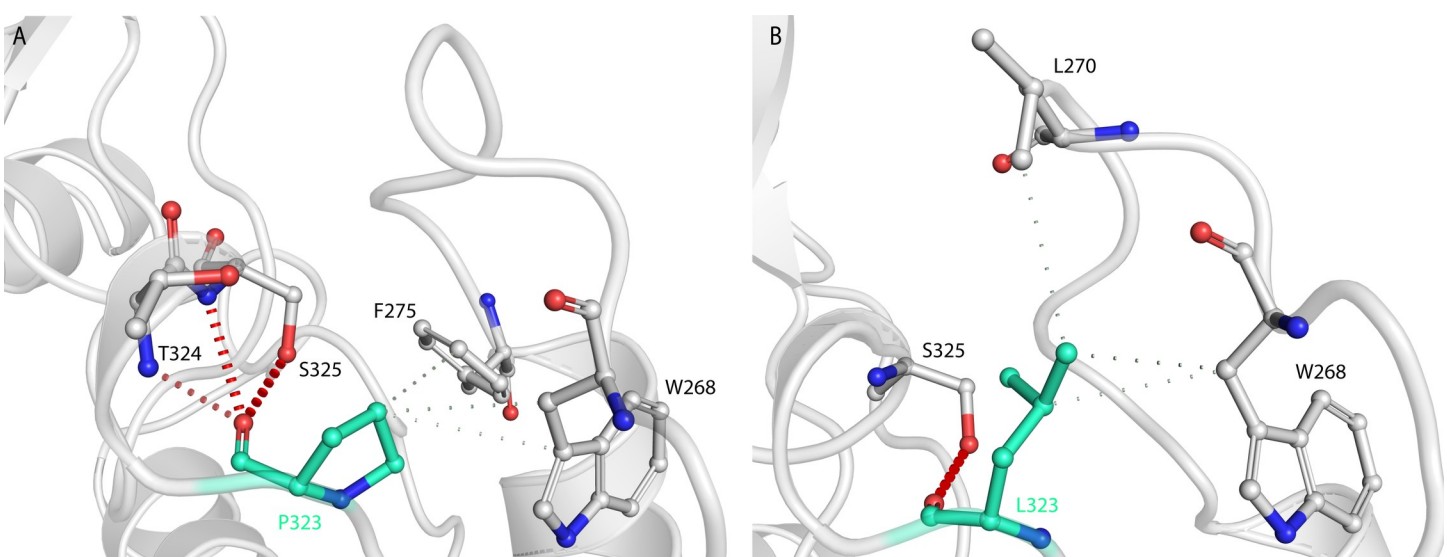

**Fig 10. 3D modelling of P323L mutation.** (A) Suggested bonding network of P323 where the COO- group might form H-bonds with the backbone NH group of T324 and S325 and the side chain of S325. The grey dashed lines depict the hydrophobic interactions between P323 and W268 and F275. (B) The mutated L323 forms a H-bond with the side chain of S325 and forms a hydrophobic interaction with L270, which is at the curve of the loop making that region more compact.

number of clusters of viral strains in different continents: Europe with six clusters, South America with five, North America with four, and Africa and Asia with two each. Both strong and weak LD blocks were seen among clusters of strains in every continent; the four dominant strains in Europe have significant proportion of strong and weak LD signatures in them. Proportions of strains carrying missense variants over synonymous variants differ among continents–five clusters of missense variants were dominant in Europe, three in North America while two clusters of missense variants were dominant in Asia, Africa, and South America. Regarding recombination patterns, European and North American continents showed evidence for the presence of recombination events among SARS-CoV-2 genomes indicating continuous evolution amid SARS-CoV-2 viral strains.

Admixture analyses have shown 7 different strains with differential segregation of alleles in SARS-CoV-2 isolates. Upon constricting variants with strong LD, the proportional assignment did not change in Africa and Asia, whereas it changed in Europe, North America, Oceania, and South America. In fact, proportions in Europe (C2 & C3) and North America (C2 & C4) increased excessively suggesting that strong LD sites were present in more than one strain in each of these two continents. The presence of an un-admixed block pattern of strong LD between strains suggests that LD sites were not broken by significant recombination seen in these two continents. This is either due to their physical distance or natural selection. On the contrary, weak LD sites have shown clear admixture between strains. Strikingly, each continent is dominated by a different set of nonsynonymous clusters such as Africa by C4, Asia by C1, Europe by C2, North America by C3, Oceania by C5, and South America by C2. This is also evident from the allele frequency variation seen in each continent.

Further, estimation of continental-wise haplotype block enabled us to identify variations in linked nonsynonymous and synonymous sites in Asia, Europe, and North America. Although selection primarily acts on variation that undergoes amino acid change, many synonymous variants were observed in haplotype blocks. This suggests that these synonymous sites hitch-hiked along with nonsynonymous variants due to their physical proximity. Another interesting feature we observed was the variation in the number of nonsynonymous sites between

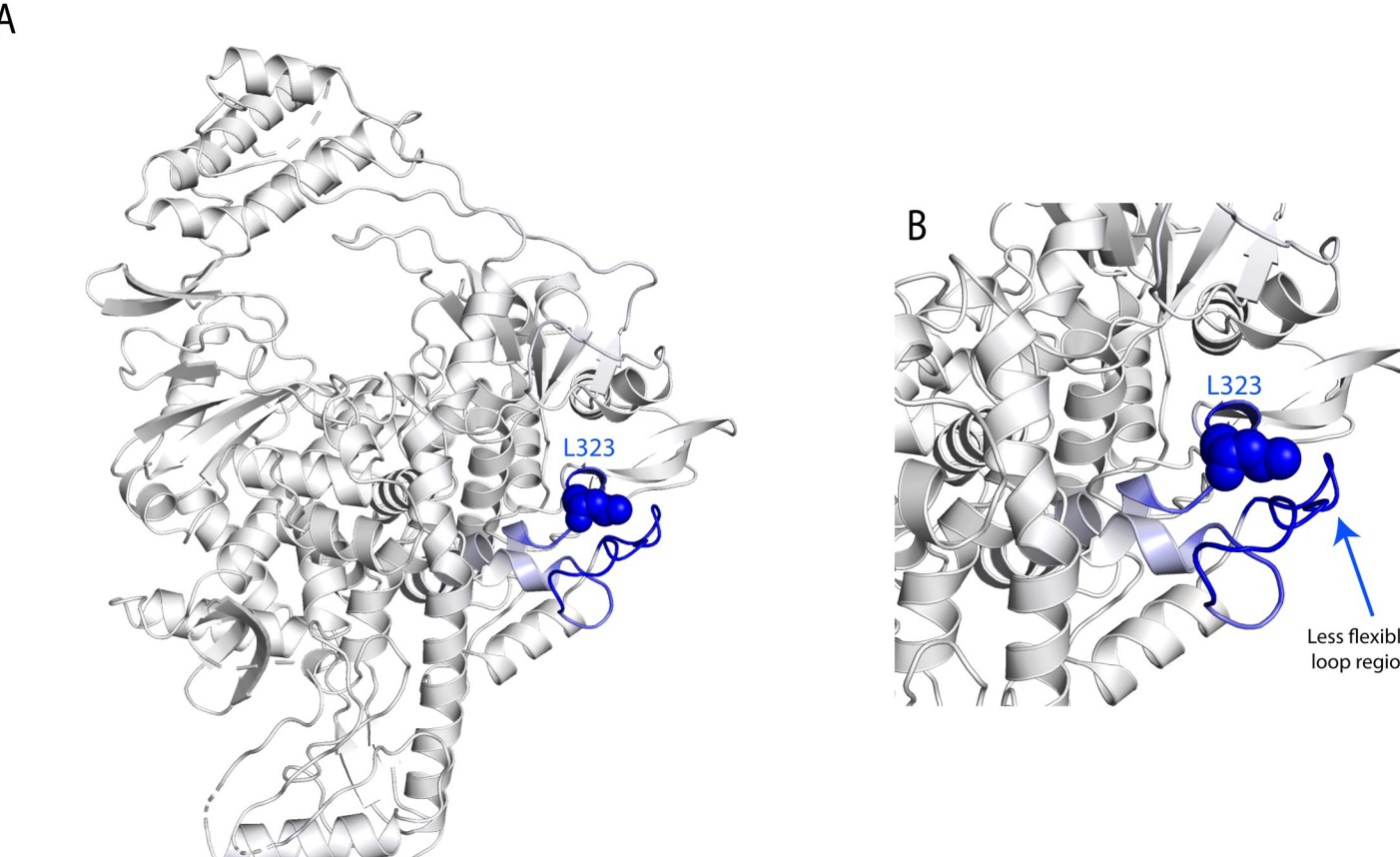

**Fig 11. 3D depiction of the less relaxed loop caused by L323 mutation.** (A) 3D structure of the RNA-dependent RNA polymerase where the blue region depicts a more rigid structure due to P323L mutation with an increase in stability of $\Delta\Delta G$: 0.717 kcal/mol and a decrease in vibrational entropy to $\Delta\Delta SVib$ ENCoM: -0.301 kcal.$mol^{-1}.K^{-1}$. (B) A zoomed-in structure showing the less flexible loop region caused by the tight hydrophobic interactions between L323 and the hydrophobic moieties.

Asia, Europe, and North America. The Asian haplotype block carried three nonsynonymous variants, European haplotype block carried ten nonsynonymous variants, and North American haplotype block carried seven nonsynonymous variants. This suggests that the initial strain, which originated and travelled from Asia, had less functional sites whereas coinfection-led recombination in Europe and North America enriched functional sites in strains.

The structural analysis of the two European strain mutations, D614G located in the spike gene and P314L located in the RdRp gene, indicated that the former mutation may render the furin cleavage site more accessible while the latter would increase protein stability. Around 73% of the European samples have both mutations segregating together; while in Africa, only 11% of the viral samples harbor these mutations. The European strain carries additional mutations, notably the hotspot mutations R203K and G204R, that cluster in a serine-rich linker region at the RdRp. It was suggested that these mutations might potentially enhance RNA binding and replication and may alter the response to serine phosphorylation events [40], which might further exacerbate SARS-CoV-2 virulence.

We understand that other confounding factors such as SARS-CoV-2 testing, socioeconomic status, the availability of suitable medical services, and the burden of other diseases are important contributors to the disparities seen in mortality rates around the world. Nonetheless, it is imperative that more functional studies are conducted to delineate the impact of these variants on SARS-CoV-2 transmissibility, diagnostics, vaccines, and therapeutics. Finally, our data

highlight the urgent need to correlate patients' medical/infection history with viral variants to anticipate the effect of these strains on the COVID-19 pandemic.

## Supporting information

**S1 Fig. Principal Component Analysis for the 2352 SARS-CoV-2 sequences distributed by their collection month.** Principal Component Analysis (PCA) blot based on the collection month (January, February, March) for the 2352 SARS-CoV-2 sequences extracted from GISAID data; PC1 (EV = 9.7030), PC2 (EV = 8.84814). Red circle indicates the founder strain that was sequenced in January.
(PDF)

**S2 Fig. Trend of CV error in the raw dataset and in variants filtered for MAF ≥ 0.5%.** The CV for the MAF>0.5% dataset comprised 72 variants shown. K = 7 is the best fit (upon observing consistency in CV error between the raw set of variants and the set of variants with MAF≥0.5% at K = 7, optimum number of clusters 7 was selected; the inconsistency observed in the raw dataset from K = 8 may be resulting from MAF<0.5% variants), suggesting that 7 different SARS-CoV-2 strains existed in early transmission of SARS-CoV-2 across continents. (CV-cross validation procedure).
(PDF)

**S3 Fig. Linkage disequilibrium (LD) variation in haplotype block of combined dataset in each continental dataset.** Extent of LD variation observed in each continental dataset when haplotype block comprising the set of 18 variants identified in combined dataset were mapped to continental datasets.
(PDF)

**S1 File. Alignments used for the recombination tests.**
(ALN)

## Acknowledgments

We gratefully acknowledge the researchers from the originating and submitting laboratories of the viral genome sequences to GISAID's EpiFlu™ Database on which this study is based. The data submitters to GISAID can be contacted directly via www.gisaid.org.

## Author Contributions

**Conceptualization:** Dania Haddad, Ashraf Al Madhoun, Thangavel Alphonse Thanaraj, Fahd Al-Mulla.

**Data curation:** Dania Haddad, Sumi Elsa John, Anwar Mohammad, Maha M. Hammad, Prashantha Hebbar, Arshad Channanath, Rasheeba Nizam, Sarah Al-Qabandi, Abdullah Alshukry, Hamad Ali, Thangavel Alphonse Thanaraj, Fahd Al-Mulla.

**Formal analysis:** Dania Haddad, Sumi Elsa John, Anwar Mohammad, Maha M. Hammad, Prashantha Hebbar, Arshad Channanath, Rasheeba Nizam, Thangavel Alphonse Thanaraj, Fahd Al-Mulla.

**Funding acquisition:** Fahd Al-Mulla.

**Investigation:** Dania Haddad, Maha M. Hammad, Ashraf Al Madhoun, Fahd Al-Mulla.

**Methodology:** Sumi Elsa John, Anwar Mohammad, Prashantha Hebbar, Arshad Channanath, Thangavel Alphonse Thanaraj.

**Project administration:** Fahd Al-Mulla.

**Resources:** Sarah Al-Qabandi, Abdullah Alshukry, Hamad Ali, Fahd Al-Mulla.

**Software:** Sumi Elsa John, Prashantha Hebbar, Arshad Channanath, Rasheeba Nizam.

**Supervision:** Ashraf Al Madhoun, Thangavel Alphonse Thanaraj, Fahd Al-Mulla.

**Writing – original draft:** Dania Haddad, Sumi Elsa John, Anwar Mohammad, Maha M. Hammad, Thangavel Alphonse Thanaraj, Fahd Al-Mulla.

**Writing – review & editing:** Dania Haddad, Sumi Elsa John, Anwar Mohammad, Maha M. Hammad, Prashantha Hebbar, Arshad Channanath, Rasheeba Nizam, Sarah Al-Qabandi, Ashraf Al Madhoun, Abdullah Alshukry, Hamad Ali, Thangavel Alphonse Thanaraj, Fahd Al-Mulla.

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
