## [Decision Letter · Decision Letter 0]

3 Mar 2021

PONE-D-20-33456

SARS-CoV-2: Proof of recombination between strains and emergence of possibly more virulent ones

PLOS ONE

Dear Dr. Hammad,

Thank you for submitting your manuscript to PLOS ONE. After careful consideration, we feel that it has merit but does not fully meet PLOS ONE’s publication criteria as it currently stands. Therefore, we invite you to submit a revised version of the manuscript that addresses the points raised during the review process.

The manuscript was reviewed by two experts. One of the reviewers recommended a minor revision, while the other recommended rejecting the manuscript. Having read the manuscript myself, I would ask you to take into account all comments which the reviewers made when revising your manuscript. Reviewer 2 made detailed suggestions to improve the presentation of recombination analyses.

We look forward to receiving your revised manuscript.

Kind regards,

Houssam Attoui, PharmD, PhD

Academic Editor

PLOS ONE

Journal Requirements:

Reviewers' comments:

Reviewer's Responses to Questions

**Comments to the Author**

1. Is the manuscript technically sound, and do the data support the conclusions?

Reviewer #1: Yes

Reviewer #2: No

2. Has the statistical analysis been performed appropriately and rigorously? 

Reviewer #1: Yes

Reviewer #2: No

3. Have the authors made all data underlying the findings in their manuscript fully available?

Reviewer #1: Yes

Reviewer #2: No

4. Is the manuscript presented in an intelligible fashion and written in standard English?

Reviewer #1: Yes

Reviewer #2: Yes

5. Review Comments to the Author

Reviewer #1: Hammad and co-workers analyzed early spread SARS-CoV-2 using sequences from GISAID.

The work is important and compelling. They should make a few changes:

Title: “SARS-CoV-2: Proof of recombination between strains and emergence of possibly more virulent ones” should be modified. Suggest: SARS-CoV-2: Possible recombination and emergence of potentially more virulent strains. The analysis does not prove recombination.

The authors should specify the gene for each mutation, at least the first time it is used. For example: Spike_ D614G and Nsp12_P314L.

Specific changes:

Line 34 P314L which is a RdRp mutation likely does not “enhance viral entry and stability.”

Line 58 Change Beta-coronaviruses to betacoronaviruses

Line 220 “cleavage site (Fig 5 C) shows a high surface accessible region, where the viral protein can attach to the host protein.” This should be clarified. The host protein I believe they are discussing is furin (or a furin-like protease) . Not clear that this is attachment pers e, which is usually reserved for the ACE2 interaction.

Line 226 and elsewhere change FURIN to furin.

Reviewer #2: The authors claimed that they had proof of recombination between SARS-CoV-2 strains and therefore, the emergence of more virulent ones. The preliminary analysis of recombination is not followed by an in a deep analysis of this process but instead, is filled with another analysis which confounds the reader for the manuscript's main message.

Regarding the mere analysis of recombination with PhiPack, which implements NSS, MaxChi, and Phi test, the authors said they found evidence of recombination for the European and American datasets. The evidence of recombination of European recombination only stands for a statistical test, with the rest of the tests largely rejecting the hypothesis. Recombination should be supported by more than one method (see Posada and Crandall 2001; doi:10.1073/pnas.241370698). For the North America dataset, the recombination signal should be analysed deeper. First, ¿how many permutations did you run? At least 100 permutations should be done. It can be seen that MaxChi2 and Phi (with unknown permutations, data not given) almost reject the recombination hypothesis. From Bruen et al., "Max χ2 and NSS falsely infer the presence of recombination under a simple model of mutation rate correlation" (doi:10.1534/genetics.105.048975). Thus, authors need to corroborate further these results. For example, they can try the program 'profile' implemented in the PhiPack package, to see with a window approach which regions exhibit the strongest evidence of mosaicism and analysed them. Which variants would be affected by recombination? Are those variants the ones further analysed in this work? Other methods can be also included. For instance, the authors determine the linkage disequilibrium (LD) but they did not correlate with the genetic distance, which is the test for recombination. As the authors do not provide enough evidence of recombination, the message of this work should be changed. Finally, alignments use for test recombination should be made accessible in case authors proved that.

6. PLOS authors have the option to publish the peer review history of their article (what does this mean?). If published, this will include your full peer review and any attached files.

Reviewer #1: No

Reviewer #2: **Yes: **Beatriz Beamud

---

## [Author Response · Author response to Decision Letter 0]

13 Apr 2021

Dear Dr. Attoui,

We would like to sincerely thank you and the reviewers for your valuable time and useful contributions. We appreciate your input which helped improving our manuscript. As suggested, we considered all comments that the reviewers made during the revision process specially to improve the presentation of our recombination analyses. Attached with the revised submission, you will find our point-by-point response to the referees and the changes we made based on the reviewers’ comments.

---

## [Editor Report · Decision Letter 1]

26 Apr 2021

SARS-CoV-2: Possible recombination and emergence of potentially more virulent strains.

PONE-D-20-33456R1

Dear Dr. Hammad,

We’re pleased to inform you that your manuscript has been judged scientifically suitable for publication and will be formally accepted for publication once it meets all outstanding technical requirements.

Kind regards,

Houssam Attoui, PharmD, PhD

Academic Editor

PLOS ONE
---

## [Editor Report · Acceptance letter]

17 May 2021

PONE-D-20-33456R1 

SARS-CoV-2: Possible recombination and emergence of potentially more virulent strains 

Dear Dr. Thanaraj:

I'm pleased to inform you that your manuscript has been deemed suitable for publication in PLOS ONE. Congratulations! Your manuscript is now with our production department. 

Kind regards, 

on behalf of

Dr. Houssam Attoui 

Academic Editor

PLOS ONE